# The Analysis of Transcriptomes and Microorganisms Reveals Differences between the Intestinal Segments of Guinea Pigs

**DOI:** 10.3390/ani12212925

**Published:** 2022-10-25

**Authors:** Chuang Tang, Jideng Ma, Fanli Kong, Bo Li, Qinjiao Du, Yali Zhang, Haoming Wang, Qianzi Tang, Silu Hu, Lingyan Liu, Xuewei Li, Mingzhou Li

**Affiliations:** 1Institute of Animal Genetics and Breeding, College of Animal Science and Technology, Sichuan Agricultural University, Chengdu 611130, China; 2Chongqing Academy of Animal Sciences, Chongqing 402460, China; 3College of Life Science, Sichuan Agricultural University, Chengdu 611130, China

**Keywords:** guinea pig, intestine, transcriptomes, microorganisms

## Abstract

**Simple Summary:**

Functional differences between the various segments of the intestine and differences in the intestines between species have been demonstrated in previous studies. Guinea pigs are commonly used model animals in intestinal-related research. As well as modeling disease states, it is important to study the intestinal tract of guinea pigs to derive basic data for the rational use of guinea pig models. Therefore, we collected six different segments of the intestinal tissue of guinea pigs and analyzed the gene expression profiles and microbial composition of each of these regions. The results revealed the functional regionalization of the intestine, with changes in functional gene expression between different intestinal segments, an association between microbial abundance and gene expression, and differences in intestinal gene expression between different species. Our study provides a reference for future intestinal-related research.

**Abstract:**

The intestine is a tubular organ with multiple functions such as digestion absorption and immunity, but the functions of each intestinal segments are different. Intestinal regionalization is necessary for normal physiological function, but it also means the research results obtained at specific sites may not be applicable to other intestinal segments. In order to comprehensively describe the functional changes in the intestine, different intestinal segments and their contents (duodenum, jejunum, ileum, cecum, colon, and rectum) of guinea pigs were collected for RNA seq and 16S rRNA seq, respectively. The results showed differential genes of each intestinal segment mainly involve mucosa, digestion, absorption, and immunity. The gene sets related to fat, bill salts, vitamins, aggregates, amino acids, and water absorption were highly expressed in the small intestine, and the gene sets related to metal ions, nucleotides, and SCFAs were highly expressed in the large intestine. In terms of immunity, the CD8^+^ T, Th1, eosinophils, pDCs, and natural killer (NK) T cells in the small intestine showed higher scores than those in the large intestine, while the pattern-recognition receptor signaling pathway-related genes are highly expressed in the large intestine. In terms of microbial composition, *Proteobacteria* and *Actinobacteria* are abundant in the small intestine, while *Firmicutes* and *Spirochaete* are abundant in large intestine. The correlation analysis showed a high correlation between intestinal microorganisms and gene modules related to digestion and absorption. In addition, cross-species analysis showed the SCFA metabolism gene expression trends in human and rodent intestine were different. In conclusion, we analyzed the changes in substance transport, immune and microbial composition between different intestinal segments of guinea pigs, and explored the relationship between intestinal transcriptome and microorganisms, our research will provides a reference for subsequent intestinal-related research.

## 1. Introduction

The intestine is mainly responsible for absorption, immunity, and microbial defense. Although the intestine is a continuous tube, the different intestinal segments exhibit distinct anatomy and functions [1,2]. The anatomy of the intestinal segment is appropriate to its function, and the small intestine is the primary site for the digestion and absorption of nutrients. The small intestine is characterized by the finger-like structures of the mucosa, which significantly increase the absorption area of the small intestine [3]. The mucosal surface of the large intestine is flat and lacks villi, and is instead covered with thick mucus [4,5]. The digestive and absorptive functions of the large intestine are weak, but it is the primary site of microbial digestion [6,7].

The gut is also considered an immune organ. There are differences in exposure to foodborne antigens and intestinal microorganisms in different intestinal segments [3]. In the proximal small intestine, there is a high nutrient concentration and low microbial abundance. The proximal intestinal immune system needs to maintain a normal intestinal barrier while allowing for absorption, and this is achieved by rapid absorption, an acidic and oxygen-rich environment, a high concentration of IgA, and a large quantity of antimicrobial peptides [1]. The nutrient concentration in the distal small intestine is low and the number of microbes increases rapidly in this region. A characteristic of the distal intestinal immune system is the abundant enrichment of Peyer’s patches (PP), which are an initiator of adaptive immunity [8]. A large number of PP and goblet cells in the distal small intestine suggests stronger antigen presentation [1]. The large intestine transports fewer nutrients, but it contains more abundant microorganisms. To protect against microbial invasion, the large intestine forms a complete physical barrier consisting mainly of dense mucus [9,10].

Intestinal regionalization is influenced by intrinsic and environmental factors [3]. However, the relationship between these factors (such as gene expression, foods, and microorganisms) and their contribution to intestinal regionalization is unclear. Intestinal microorganisms colonize from birth to adulthood, *Enterobacter* and *bifidobacteria* are early bacteria colonized, the structure of intestinal microorganisms tends to be stable in adulthood [6,11]. The interaction between microorganisms and intestine will affect the characteristics of intestinal microorganisms and the functions of intestine. For example, Compared with the microorganisms in luminal contents. the *Bacteroides thetaiotaomicron* and *Escherichiacoli* colonized in the colonic mucous layer show different transcriptional profiles [12]. The *B. The taiotaomicron* and *Faecalibacterium prausnitzii* can enhance the expression of genes related to mucin glycosylation, thereby regulating the production of mucus [12]. Studies on germ-free animals also shown that intestinal microbes can regulate gene expression in the host, such as phase I enzymes, phase II enzymes, transporters, and transcription factors [13,14]. Meanwhile, microbial colonization is not random and is influenced by the host gene expression. For example, the *MyD88* is an important adapter molecule for intestine to recognize microorganisms, deletion of the *Myd88* gene resulted in higher microbial diversity and enrichment of filamentous bacterial fragments in the small intestine of mice [15]. There are also variations in the composition and function of intestinal microorganisms among species [16]. Although the main intestinal microorganisms of humans and guinea pigs are both *Bacteroidetes* and *Firmicutes*, guinea pigs have a higher abundance of intestinal microbes, such as the mucin-degrading *Akkermansia* and the methanogenic archaea *Methanobrevibacte*. In addition, the intestinal microbes of guinea pigs seem to be more involved in the dehydration/rehydration stress pathways [17].

The guinea pig is a commonly used model animal for intestinal studies, but there have been fewer reports on the guinea pig intestine than on the intestines of mice and rats [18]. Differences in the gut between species may affect the accuracy of experiments, such as preclinical trials of oral drugs [19]. It is therefore necessary to carry out a detailed study of the gut of guinea pigs, considering the influence of differences between species on experimental studies. Our study describes the full-length intestinal transcription profile and microbial composition of normal guinea pigs, providing reference data for intestinal research and the rational use of guinea pigs as an experimental model.

## 2. Materials and Methods

### 2.1. Ethics Statement

All research involving animals was conducted according to Regulations for the Administration of Affairs Concerning Experimental Animals (Ministry of Science and Technology, Beijing, China, revised in March 2017) and approved by the animal ethics and welfare committee (AEWC) of Sichuan Agricultural University under permit No. DKY-B2019202011. This study was carried out in compliance with the ARRIVE guidelines.

### 2.2. Animals Materials

Six male Guinea pigs (China Science and Technology Resource: 15497.09.NR030200001, 7–8 weeks old) were purchased from Chengdu Dossy Experimental Animals Co., Ltd. (Chengdu, China). All Guinea pigs were fed with standardized diet (provided by Chengdu Dossy) in the laboratory. The animals were dissected quickly after euthanasia, and the junction of each intestinal segment was ligated with cotton thread to prevent the content from flowing. Collect samples from duodenum to rectum section by section, gently squeeze the intestinal contents into the cryopreservation tube with tweezers, then cut off the current intestinal tract, flush it with PBS buffer and put it into the cryopreservation tube. The tissues and contents of each intestinal segment were collected, frozen in liquid nitrogen, then stored at −80 °C.

### 2.3. Total RNA Extraction, Library Preparation and Sequencing

Total RNA was isolated from the 26 tissue samples using the HiPure Total RNA Mini Kit (Magen, Guangzhou, China) according to the manufacturer’s instructions. The integrity and quality of total RNA samples were analyzed with NanoDrop 2000 (Thermo Fisher Scientific, Wilmington, DE, USA) and Bioanalyzer 2100 system (Agilent Technologies, Palo Alto, CA, USA). The RNAs with a ratio of absorbance at 260/280 nm ranging from 1.8 to 2.0 and RIN value > 1.8 were selected for further study. Qualified samples were constructed for Library in Novogene (Beijing, China), and high-throughput sequencing was performed on the DNBSEQ-T7 platform. The low-quality reads were removed, namely, those with ≥10% unidentified nucleotides, >10 nt aligned to the adapter, and with >50% of bases with Phred quality < 5.

### 2.4. Quantification of PCGs and lncRNAs

The clean data were mapped to the Cavia porcellus Cavpor 3.0 reference genome using STAR (v.2.6.0c), the STAR parameter we used is “STAR --runThreadN 15 --genomeDir ${genomeDir} --readFilesCommand zcat --outSAMtype BAM Unsorted SortedByCoordinate --outFilterType BySJout --outFilterMultimapNmax 20 --alignSJoverhangMin 8 --alignSJDBoverhangMin 1 --outFilterMismatchNmax 999 --outFilterMismatchNoverReadLmax 0.04 --alignIntronMin 20 --alignIntronMax 1000000 --alignMatesGapMax 1000000 --chimSegmentMin 10 --outFilterIntronMotifs RemoveNoncanonical --outSAMstrandField intronMotif --outReadsUnmapped Fastx --readFilesIn ${fq1} ${fq2}”. Protein coding genes (PCGs) and long non-coding RNAs (lncRNAs) annotation information was extracted from Cavia_porcellus.Cavpor3.0.103.gtf and Cavia_porcellus.Cavpor3.0.dna.toplevel.fa. The transcripts per kilobase million (TPM) of PCGs and lncRNAs were calculated by Kallisto (v0.44.0), the parameters of Kallisto use default values. We considering the expression characteristics of different transcripts, the PCGs with TPM > 1 in at least three samples were defined as expressed PCGs (lncRNAs with TPM > 0.1 in at least three samples) [20].

### 2.5. Tissue Specific and Alternative Splicing Analysis

The tissue specificity index (TSI) was used to evaluate the tissue-specific expression level of genes. The range of TSI was 0~1, and higher TSI indicates the gene showed higher tissue specificity [21]. The Cassette exons may have variable shear events with adjacent exons, the ratio between reads including or excluding exons, was called percent spliced in index (PSI), which was used to quantify alternative splicing [22]. Based on the BAM file obtained from STAR, the number of reads on the exon is obtained by using JavrKit biostar103303 software. PSI = 0 or PSI = 1 means there was no alternative splicing. PSI = 0 means that the exon is completely skipped, and PSI = 1 means that it is not skipped. Therefore, only exons with a PSI of 0.05~0.95 and existing in at least three samples were retained.

### 2.6. Differentially Expressed Genes Analysis and Functional Enrichments

The differentially expressed genes (DEGs) were calculated by edgeR (Bioconductor version: Release 3.10). The genes with |fold change| > 2 and *p* value < 0.05 were considered to be DEGs. The DEGs functions enrichment, which includes Gene Ontology (GO) and Kyoto Encyclopedia of Genes and Genomes (KEGG) were performed at the Metascape web portal (http://metascape.org). The GO terms or KEGG pathways with Benjamini corrected *p* < 0.01 were considered to be significant.

### 2.7. Gene Set and Cell Composition Analysis

The statistical analysis was performed using R (version 4.0.3). The gene sets related to sugar, lipid, amino acid, bile salt, vitamin, organic solutes, metal ion, nucleotide, fungus, and bacterium came from published literature [5]. The gene sets related to water, SCFAs, PRRs pathway, and mucus secretion came from the quick GO database [23]. Since the gene sets was mainly derived from mice or human studies, the genes that are only one-to-one orthologs with guinea pig were remained. The homologous gene information was downloaded from the Ensembl database (http://feb2021.archive.ensembl.org/biomart/martview/, access date 1 October 2022). The trend of genes was represented by standardized TPM. The cell composition was evaluated using the cell score calculated by Xcell [24].

Human RNA-seq data comes from HPA and GTEx (https://www.prteinalas.org/download/rna_tissue_consensus.tsv.zip, access date 1 October 2022), mouse and rat RNA-seq data came from EBI ArrayExpress (data accession: E-MTAB-6081) [25]. Because human data treat the jejunum and ileum as the same group, other species also adopt the same grouping method to ensure comparability. The genes that are one-to-one orthologs to human and exist in all species are retained, and the analysis of gene expression trend is the same as in previous steps.

### 2.8. DNA Isolation, Library Preparation and Sequencing

The DNA of intestinal microorganisms was isolated using the E.Z.N.A Stool DNA Kit (Omega Bio-tek, Inc., Norcross, GA, USA) according to the manufacturer’s instructions. The V3–V4 regions of the bacterial 16S rRNA gene were amplified with the Primer pairs (forward: CCTAYGGGRBGCASCAG, reward: GGACTACNNGGGTATCTAAT), and sequenced on the Illumina NovaSeq 6000 platform by Novogene (Beijing, China). The 16S rRNA seq data were analyzed using QIIME2 (version 2020.8). In brief, the DADA2 pipeline of QIIME2 was used to Filtering data and generate absolute sequence variables (ASV) table. The alpha-diversity indices and beta-diversity indices based on ASV were calculated by QIIME2. The principal coordinates analysis (PcoA) based on the distance matrix was performed by the ape R package (version 5.5). The primer sequences 347F and 803R were used to extract the V3-V4 information of the SILVA database release 132. The classifier, trained with the extracted data, is used to annotate species information.

### 2.9. Weighted Gene Correlation Network Analysis

The Weighted correlation network analysis was completed by the WGCNA R package (version 1.70-3). The Soft threshold was set to 9 according to the results of PickSoftThreshold function. The coexpression network was obtained based on Pearson correlation, the topological overlap metric and clustering gene modules are calculated by blockwiseModules function with mergeCutHeight of 0.25, and other parameter setting refers to the previous study [26]. The module eigengene (ME), which was essentially the first principal component of the module, was usually used to represent the characteristics of the module. Hmisc R package (version 1.70-3) was applied to calculate correlations between ME and Microbial abundance. The combination with *p* < 0.01 and |r| > 0.80 was considered to be highly correlated.

## 3. Results

### 3.1. Comparison of the Transcription Profiles of the Different Intestinal Segments of Guinea Pigs

A total of 169 Gb of clean RNA-seq data were obtained after removing low-quality reads. The clean data were mapped to the *Cavia porcellus* Cavpor 3.0 reference genome with a mapping ratio of 90.28–96.97% (Appendix A). In total, 11,969 expressed (TPM > 1 in at least three samples) protein-coding genes (PCGs) and 1701 expressed (TPM > 0.1 in at least three samples) long non-coding RNAs (lncRNAs) were obtained after quantification.

Principal component analysis (PCA) based on the PCG expression profile of the samples showed that the samples were separated according to the large and small intestine on principal component 1 and according to different intestinal segments on principal component 1 (Figure 1A). Similar to the PCA results, the correlation heatmap between samples showed a low correlation between the large and small intestine (Figure 1C). We found that the correlation in the large intestine (mean r = 0.93) was lower than that in the small intestine (mean r = 0.97), suggesting that the different segments of the large intestine may have a higher degree of functional specialization (Figure 1E). In addition, the percent spliced index (PSI) was used to evaluate the alternative splicing levels of PCGs. The clustering results indicated there was also an obvious difference in alternative splicing between the large and small intestines (Appendix A). Compared with PCGs (median TPM value = 2.63), lncRNAs showed a lower expression level (mean TPM value = 0.45) (Figure 1F). The clustering results obtained for the lncRNAs expression profile were similar to those for the PCGs (Figure 1B), but the correlation of the lncRNAs expression profile (r = 0.71~0.96, median = 0.83) between different intestinal segments was lower than that of mRNAs (r = 0.78~0.99, median = 0.92), which may be because of the higher tissue specificity of lncRNAs (Figure 1D). Next, we used the tissue-specific index (TSI) to evaluate the tissue-specific expression levels of genes. Consistent with the conjecture, lncRNAs showed higher tissue-specific expression levels than PCGs (Figure 1G).

### 3.2. Functional Differences between Intestinal Segments

To further investigate functional differences between segments of the intestine, we used EdgeR to identify the differentially expressed genes (DEGs) (*p* < 0.05, |fold change| > 2) between each pair of segments (Figure 2A). We used the DEGs shared by nine combinations (small intestine vs. large intestine) to represent the common differences between small and large intestine. The enrichment terms of the DEGs reflected the well-known differences between segments, including the structure of the intestinal mucosa, digestion and absorption, and immune function (Figure 2B).

Among the top 20 enrichment terms ranked by *p* value, 9 enrichment terms were shared by all three combinations of the small intestine, and these involved the transport of substances, signaling receptor activity, and secretion (Figure 2D). Previous studies suggested functional differences between the proximal and distal portions of the small intestine [27]. In all combinations of the small intestine, the number of DEGs between the duodenum and ileum was the largest. Moreover, there were significant differences in lipid metabolism between the ileum and other small intestinal segments. In the large intestine, there were 14 enrichment terms shared by all three combinations of the large intestine (Figure 2D). Although there were more DEGs between segments of the large intestine (mean = 1267) than the small intestine (mean = 215), similarly to the small intestine, the main functions of the DEGs of the large intestine included the transport of substances, neural signaling, and secretion (Figure 2F). The differences in transport in the small intestines mainly involved fats (GO:0010876: lipid localization, mmu04975: Fat digestion and absorption), amino acids (GO:1901605: alpha-amino acid metabolic process, GO:0015849: organic acid transport), and carbohydrates (GO:0009743: response to carbohydrate, GO:0019752: carboxylic acid metabolic process), and the differences in transport in the large intestines involved ions (GO:0006820: anion transport, GO:0043269: regulation of ion transport, GO:0030001: metal ion transport). In addition, the cecum differs from other segments of the large intestine in terms of carboxylic acid metabolism (GO:0019752: carboxylic acid metabolic process) and morphogenesis (GO:0032989: cellular component morphogenesis). In conclusion, differences in digestion and absorption exist between all segments of the intestine, and some intestinal segments may have a preference for the metabolism of specific nutrients; for example, the cecum may play a critical role in the metabolism of SCFAs (GO:0019752: carboxylic acid metabolic process). By contrast, differences in immunity mainly exist between the small intestine and the large intestine, especially regarding leukocyte differentiation (GO:0070663: regulation of leukocyte proliferation) (Figure 2B).

### 3.3. Distinct Expression Patterns of the Functional Gene set in the Intestines

Enrichment analysis of DEGs showed differences in absorption, and immune and secretory functions between intestinal segments. To better understand the functional changes in intestinal segments, we collected relevant gene sets to investigate their expression trends in the gut. The nutrient absorption gene sets were involved in sugar, lipid, amino acid, bile salt, vitamin, water, organic solute, SCFA, metal ion, and nucleotide absorption. In general, the gene sets related to fat, bile salts, vitamins, sugars, amino acids, and water absorption were highly expressed in the small intestine. The gene sets related to metal ions, nucleotides, and SCFAs were highly expressed in the large intestine. The genes related to organic solvent absorption were similarly expressed in all intestinal segments. Although the primary absorption of nutrients occurs in the small intestine, the high expression of some transport genes in the large intestine suggests that the large intestine may be involved in the transport of specific substances, such as SCFAs, metal ion, and nucleotide (Figure 3A). Then, we studied the expression pattern of single transporter gene. Surprisingly, contrary to the overall expression trend of gene sets, some transporter genes were high expressed in the large intestine rather than the small intestine, such as the genes related to sugars (*SLC2A10*, *SLC50A1*), lipid (*FABP2*, *ACAT2*, *HMGCR*), amino acids (*SLC38A2*, *SLC25A12*, *SLC1A5*, *SLC25A13*), vitamins (*CD320*, *BTD*, *RDH5*), and water (*AQP8*) (Figure 3B). The gene counts table and TPM table are provided in the Appendix A (Appendix A).

Although different intestinal segments can exhibit similar immune functions, the composition and activity of immune cells still show regionalization [3]. DEG enrichment data suggest that there are differences in leukocyte differentiation between the large intestine and the small intestine (regulation of leukocyte proliferation, GO: 0070663, *p* = 7.28 × 10^−8^), so we focused on the composition of intestinal immune cells. Cell composition analysis of samples was conducted using xCell [24]. The enrichment scores for lymphocyte cells (CD8^+^ T cells and Th1 cells) and non-lymphoid innate immune cells (eosinophils, pDCs, and natural killer (NK) T cells) in the small intestine were higher than those in the large intestine, and macrophages showed high scores in the cecum (Figure 3C) [3]. Microorganisms are an important factor affecting the regional specialization of the intestinal immune system. We investigated the expression patterns of microbial-related gene sets, including the response to bacteria and fungi, the pattern recognition receptor (PRR) signaling pathway, and mucus secretion (Figure 3C). The genes related to bacterial response are highly expressed in the small intestine, while the genes related to fungal response are mainly expressed in the large intestine, which may indicate that the intestinal segment has different defense capabilities against different microorganisms (Figure 3D) [28]. The pattern-recognition receptor (PRR) signaling pathway genes are highly expressed in the large intestine, including Toll-like receptors (TLRs), NOD-like receptors, and *MyD88* (Figure 3D). The mucus layer is an important barrier within the large intestine. Genes involved in mucus secretion are highly expressed in the colon and rectum but not in the cecum.

### 3.4. Microbial Composition of the Different Intestinal Segments of Guinea Pigs

As mentioned above, microorganisms affect intestinal function, so we used 16S sequencing to study the microbial structure of the different intestinal segments. A total of 1,254,065 OTUs were obtained from 32 samples of intestinal contents, and each sample included 52,430–32,525 OTUs (Appendix A). Alpha diversity analysis showed that microbial abundance increased with the extension of the intestine, and the number of features for the large intestine was higher than for the small intestine (Figure 4A, Appendix A). Compared with those for the large intestine, the Shannon values for the small intestines of different individuals vary greatly, which may indicate that microbial composition of the small intestines is more variable. Similar to the transcriptome results, cluster analysis based on unweighted UniFrac distances showed that intestinal microorganisms were separated according to the small intestine and the large intestine on PcoA1 (Figure 4B). The main microbial species in each intestinal segment were similar at the phylum level. Six of the top ten microorganisms in each intestinal segment were the same. However, at the genus level, the microbial composition of each intestinal segment was quite different, and all segments shared only one dominant microorganism. Among the predominant microbes, *Proteobacteria* and *Actinobacteria* were significantly more abundant in the small intestine than the large intestine, whereas *Firmicutes* and *Spirochaetes* were more abundant in the large intestine (*p* value < 0.05).

To investigate the relationship between microbial abundance and gene expression, we considered microbial abundance as a phenotypic trait of the gut, utilizing WGCNA to perform an association analysis between microbial abundance and gene expression [29]. According to the expression pattern, all expressed PCGs were divided into 12 modules, and the gene enrichment results suggested that specific physiological functions may be associated with each module. Correlation analysis showed that all modules, with the exception of module 10, possessed significantly related microorganisms (Figure 4E). Taking |r| > 0.8 to indicate a high correlation, most of the high correlation combinations (16/17) were distributed in module 2. Module 2 included 616 genes, which were mainly related to nutrient transport and metabolism (Figure 4F). Most microorganisms highly associated with module 2 have been reported to be affected by diet, such as *Ruminococcus*_1, *Rikenelaceae_RC9_gut_group*, and *Tyzzerella* [30,31,32]. Other microorganisms affect intestinal function by producing SCFAs, such as *Treponema*_2, *Ruminiclostridium_9*, and *Lachnospiraceae_UCG_001* [33,34,35]. The complete module function annotation data and the correlation analysis results are provided in Appendix A (Appendix A).

### 3.5. Differences in Gene Expression Related to Absorption among Species

As a result of the differences in intestines among species, the results obtained in animal experiments may not accurately reflect the response in human intestines. For example, no significant correlation between oral drug bioavailability in rats and humans (r^2^ = 0.29) was reported [36]. To explore the differences in intestinal absorption between different species, we compared the expression patterns of digestion and absorption-related gene sets in the intestines of four different species. We only retained the one-to-one homologous genes expressed in all four species to ensure the consistency of the data. In short, all species showed similar expression patterns for lipid, amino acid, bile salt, metal ion, nucleotide, and vitamin-related gene sets (Figure 5A). The expression of human sugar transporter genes gradually decreased with the extension of the intestine, while the whole small intestines of mice, rats, and guinea pigs highly express sugar transporter genes. SCFA metabolism-related genes in mice, rats, and guinea pigs are mainly highly expressed in the large intestine, while in humans, they are mainly highly expressed in the small intestine. No apparent trend was detected in the water transport gene set.

Next, we analyzed the expression patterns of single genes (Figure 5B). Most sugar, fat, and amino acid transport genes, such as *SLC2A5*, *FABP2*, and *SLC3A2*, have similar expression patterns among species. The expression of *SLC2A2* was previously reported to show species differences, but our results did not confirm this. *FABP6* is highly expressed in the distal small intestine of different species, indicating similar bile acid transport sites between species. *SLC31A2* encodes a copper transporter, and the expression patterns of *SLC31A2* in humans and rodents are significantly different. Consistent with previous studies, *SLC31A2* is expressed at a low level in the human large intestine, while in rodents, it shows the opposite trend. As a vesicular nucleotide transporter, the expression of *SLC17A9* is essential for neurons to respond to purinergic signals, and *SLC17A9* shows different expression patterns in human and rodent intestines. *CD320* has been reported to be highly expressed in human colonic epithelial cells. It shows a similar expression pattern among species, which confirms that the large intestine is also involved in the absorption of vitamins. In the metabolism of SCFAs, significant differences in the expression of *PCK1*, *PCK2*, and ACADS between humans and rodents are evident, that may be caused by differences in the concentration of intestinal SCFAs between species. Although no obvious trends in the water transport gene set have been detected, some water transport genes, such as *AQP8* and *AQP11*, display similar expression profiles between humans and rodents.

## 4. Discussion

There have been many reports on the study of intestinal function. Most studies have noticed the differences between intestinal segments, usually dividing the intestine into the small and large intestines. However, this simple classification may not provide sufficiently detailed data for the interpretation of all experimental results [1]. Here, we studied the entire length of the intestinal tissue of guinea pigs and the intestinal microorganisms that reside within on a more detailed scale, which aids our understanding of the functional changes in different segments of the intestine.

LncRNA is widely involved in the regulation of intestinal function. *LncRNA uc.173* promotes the expression of *CLDN1* by binding to *miR-29b* and then enhances the barrier function of the intestinal epithelium [37]. LncRNA *DQ786243* can regulate the function of Treg cells by changing the expression of *CREB* and *FOXP3* [38]. The significant upregulation of lncRNA *DQ786243* in the intestines of patients with Crohn’s disease suggests that it may play an important role in pathogenesis [38]. Consistent with previous studies, intestinal lncRNAs showed low expression but high tissue specificity [20]. Therefore, when evaluating the expression of lncRNAs, we have adopted a lower standard (TPM > 0.1 in at least three samples) than PCGs (TPM > 1 in at least three samples), and we have finally retained 66% (11969/18095) of PCGs and 64% (1701/2634) of lncRNAs. Compared with PCGs, intestinal lncRNAs have greater tissue specificity, which may help to reveal the functional differences between intestinal segments. Compared with the pre-transcriptional regulation of gene expression, post-transcriptional regulation of lncRNAs is more convenient and flexible, which may be important for the timely adjustment of gene expression to cope with the changing intestinal cavity environment [39]. *SLC2A8* encodes the glucose transporter *GLUT8*. Only complete mRNA transcript was detected in the testis, while three additional transcriptional variants were found in the intestines [40]. The alternative splicing of *SLC2A8* in the intestines may reflect a regulatory mechanism of gene expression through nonsense-mediated decay [40].

Differences between the small and large intestine mainly involve the mucosal structure, digestion and absorption, and immunity. Fully differentiated intestinal epithelial cells are characterized by microvilli with a brush border structure, which increase the surface area of the intestine. However, not all microvilli are involved in absorption and secretion, and 59% of the DEGs enriched in brush border entries (GO:0005903, *p* = 3.32 × 10^˗14^, *n* = 36) belong to the SLC family, which is involved in the transport of substances [41]. Therefore, we speculate that this may indicate differences in the function of the brush border between the small and large intestines. In addition, DEGs between the small and large intestine were enriched in anchoring junction function (GO:0070161, *p* = 8.51× 10^˗0.9^, *n* = 75). The mucosa of the small intestine needs to ensure specific permeability to absorb nutrients, whereas a complete mucosal barrier is necessary for the large intestine to prevent the invasion of microorganisms, and the barrier claudins-1, 3, 4, 5, and 8 are highly expressed in the colon to enhance this barrier [42]. This functional difference is also reflected in the regulation of mucosal permeability. In mice, a high-fat diet increased intestinal permeability, which may be due to decreased tight junction protein expression in the small intestine, while no similar correlation was observed in the colon [43]. Similarly, the effect of zonula occludens toxin on intestinal permeability was limited to the small intestine [44].

Most of the substance transport genes collected in this study, including those for sugar, lipid, amino acid, bile salt, vitamin, and water, are expressed in the small intestine (Figure 3B). The large intestine may play an important role in the absorption of SCFAs, metal ions, and nucleotides (Figure 3B). Monosaccharide transport genes are highly expressed in the small intestine, such as *SLC2A5*, *SLC5A1*, *SLC5A11*, *SLC2A2*, and *SLC37A4* [26,45,46,47]. However, the glucose transport gene *SLC2A10* is highly expressed in the large intestine [48]. In addition, the high expression of *SLC50A1* suggested that the large intestine may specifically transport aldoses [26]. Most lipid transport genes were highly expressed in the small intestine, such as *APOA4*, *APOA1*, and *PLIN3* [49,50]. The cholesterol and ketone body metabolism genes, *HMGCR* and *ACAT2*, were highly expressed in the large intestine [51,52]. Neutral amino acids, basic amino acids, and excitatory amino acid transport-related genes, such as *SLC6A19*, *SLC25A39*, and *SLC1A1*, were highly expressed mainly in the small intestinal segments [53,54,55]. Conversely, some acidic amino acid and glutamine transport genes, such as *SLC25A12*, *SLC25A13*, *SLC38A2*, and *SLC1A5*, were highly expressed in the large intestine [56,57]. The ileum is the major bile acid absorption site. Consistent with this, the bile acid transporter (*SLC10A2*) and bile acid binding protein (*FABP6*) were highly expressed in the ileum [58,59]. Vitamin transport-related genes were highly expressed mainly in the small intestine, but vitamin A, H, and B12 metabolism-related genes, such as *RDH5*, *BTD*, and *CD320*, were still expressed in the large intestine [60,61,62]. Water transport genes, such as AQP10 and *AQP11*, were highly expressed predominantly in the proximal small intestine [63]. The expression of *AQP8* in the large intestine suggests that the large intestine is also involved in water transport [64]. Although the expression of the organic solute transport gene set did not show a clear trend, some of these genes, such as *SLC16A9*, *SLCO3A1*, *SLC26A6*, and *SLC51B*, still showed differences in the small and large intestine. The expression patterns of the dicarboxylic acid transporter (*SLC13A2*) and the choline transporter (*SLC44A1*) have also been reported in previous studies [26]. Among metal ions transporters, the Na, K, Ca, and Fe transporters (*SCNN1A*, *KCNS3*, *ATP2A3*, and *SLC39A8*) are highly expressed in the large intestines [65,66,67,68]. The Zn transporter (*SLC39A4*) has also been reported to be highly expressed in the small intestine, and another Zn transporter (*SLC28A2*) shows a similar expression trend [26]. Nucleotide transporter genes are mainly expressed in the large intestine, and previous studies have reported the enrichment of the centralized nucleoside transporter family (*SLC28A1*) in the small intestine, with similar results also being shown for the Zn transporter (*SLC28A2*) [26]. It is worth noting that absorption capacity of each segment of the small intestine is not consistent. The absorption of lipophilic drugs by simple diffusion mainly occurs in the proximal small intestine, while the permeability of weakly alkaline drugs with higher pKa values is higher in the distal small intestine [69]. In the large intestine, we found that the nutrient absorption capacity of the cecum was weak. However, SCFA metabolism genes are highly expressed in the cecum. *PCK1* and *PCK2* are regulatory genes in gluconeogenesis. Their expression has been proven to be positively correlated with the concentration of SCFAs [70,71]. *ACADS* can promote butyric acid metabolism, while preventing butyric acid from inhibiting the proliferation of crypt stem cells [72]. The SCFAs produced by microbial fermentation in the large intestine are important energy sources for monogastric herbivores. The high expression of SCFA metabolism genes suggests that the giant cecum of guinea pigs may be the primary site of microbial activity [73]. Finally, it should be noted that intestinal gene expression is regulated by diet. For example, high calcium diet can inhibit the expression of *ECaC2* and *ECaC1* in mouse duodenum [74]. High fructose diet can improve the fructose transport efficiency of rat intestine by increasing the expression of *GLUT5* [75]. Our results may only reflect the intestinal transcriptome of guinea pigs under laboratory conditions. When the dietary composition changes, it may be necessary to reconsider the differences between intestinal segments.

The PRR signal path is important for intestine to recognize microorganism [76]. Microorganisms may regulate the expression of TLRs. The expression of *TLR2* and TLR4 in the intestine of specific-pathogen-free mice is regionally specific, but this phenomenon is not observed in germ-free mice [77]. *NOD1* regulates neutrophil function and promotes a rapid response to infection by recognizing microbial peptidoglycans, and its expression pattern may reflect the innate immune differences between the large and small intestines [78]. The high expression of *MYD88* in the large intestine may help maintain barrier function. *MyD88* is necessary to maintain the physical distance between microorganisms and the intestine. The absence of *MyD88* facilitates easier access of microorganisms to the intestinal mucosa, and the lack of RegIIIγ, which is regulated by *MyD88*, also enables microorganisms to pass through the mucus layer more easily [79]. In addition, the small intestine and large intestine showed differences in leukocyte differentiation (GO:0070663, *p* = 7.28 × 10^˗0.8^, *n* = 39), which was also confirmed by cell score data [80]. Our results support the view that pDCs are restricted to the small intestine, and the high correlation between *CCR9* expression and the pDC score (r = 0.67) confirms that the recruitment of pDCs is CCR9-dependent [81]. Unfortunately, the annotation of the xCell database is based on human immune cells, and our data lost important cell information during homologous transformation, such as for NK cells and CD4^+^ T cells.

Intestinal microorganisms are a crucial factor in the regulation of intestinal function. The microorganisms in the large intestine and small intestine show apparent differences. The distribution of microorganisms is affected by diet. Higher proportions of *Proteobacteria* and *Actinobacteria* were detected in the intestines of individuals on high-meat and high-fat diets, while an increase in crude fiber in the diet reduced the abundance of *Actinobacteria* [82]. These characteristics make them more likely to be colonized in the small intestine with more nutrition. The ratio of *Firmicutes* to *Bacteroidetes* (F/B) is used to assess intestinal homeostasis [83]. The F/B ratio in the intestines of obese individuals is increased, while a decrease has been reported in inflammatory enteritis [83]. Our results suggested that the proportion of *Firmicutes* in the small intestine and large intestine was significantly different (*p* < 0.05). Microbial abundance in the small intestine varies significantly between individuals, so the impact of sampling location on the data should be considered when analyzing F/B ratios. *Spirochaetes* have been observed in the large intestine of guinea pigs, which are usually considered pathogenic microorganisms [84]. Compared with human intestines, guinea pig intestines are enriched in *spirochaetes*, but their role in normal intestines is unclear [15]. Intestinal microorganisms have the ability to regulate host gene expression. The gene expression of normal *HDAC3* deletion mice is uncontrolled, but this phenomenon is not observed in sterile *HDAC3* deletion mice, which indicates that intestinal microorganisms can affect the modifying enzymes of the host and thereby regulate gene expression [85]. We used WGCNA to analyze the correlation between intestinal gene expression and microbial abundance. Consistent with previous studies, our data showed a high correlation between microorganisms and nutrient digestion and the absorption gene module. Our results were suggestive of an association between microorganisms and specific gene modules, but further studies are needed to confirm an actual interaction. For example, *Lachnospiraceae_UCG_010* was highly correlated with module 4 (r = 0.82), which is involved in muscles and movement, but relevant results about *Lachnospiraceae*_UCG_010 have not previously been reported. Taken together, our findings provide some support for annotating the complex functions of microorganisms.

The intestines of different species vary greatly, which needs to be taken into consideration when using a model animal to simulate the human intestine [19]. For example, rats with a higher intestinal water content may be a better choice for solid administration experiments compared with mice. In addition to the intestinal environment, there are also species differences in the transport of substances. The sodium-dependent phosphate transporter (NaPi-IIb) encoded by *SLC34A2* is expressed in the proximal small intestine of rats and humans but not in the intestine of monkeys. Although there is also NaPi-IIb-mediated phosphate transport in mouse intestines, NaPi-IIb is mainly expressed in the distal small intestine of mice [86]. For consistency with the data on other species, we combined data on the jejunum and ileum of guinea pigs into the same group in our study. This did not change the expression patterns for most gene sets, with the exception of the vitamin gene set, which may show a significant difference in expression between the jejunum and ileum. In addition, the living environment of animals quoted in our research is difficult to trace, which makes it difficult to distinguish the differences caused by the environment from the inherent differences between species.

Finally, it should be noted that our data are all from male guinea pigs raised in the laboratory environment, which may ignore the impact of other factors on intestinal segment regionalization, such as gender, breed and living environment. The expression of intestinal transport genes and microbial composition are affected by diet, but when comparing across species, it is inappropriate to unify the diet of all species. In order to avoid the impact of inappropriate diet on the study, it is necessary to strictly control diet according to the nutritional needs of different species.

## 5. Conclusions

This study provides transcriptome and microbial data across the entire length of the intestine of guinea pigs, highlighting differences between intestinal segments that provide insight into functional changes throughout the intestine. The results showed there were significant differences in gene expression in different intestinal segments of guinea pigs, especially transport and immune genes. The cell score indicated the distribution of leukocyte population in the intestine, which would help explain the intestinal immune regionalization. We also described the dominant microorganism of different intestinal segments and analyzed the potential functions of microorganisms with transcriptome data. In addition, cross-species analysis revealed differences in intestinal transcription between humans and common model animals, which provides a reference for the rational use of model animals in future experimental studies.

## Figures and Tables

**Figure 1 animals-12-02925-f001:**
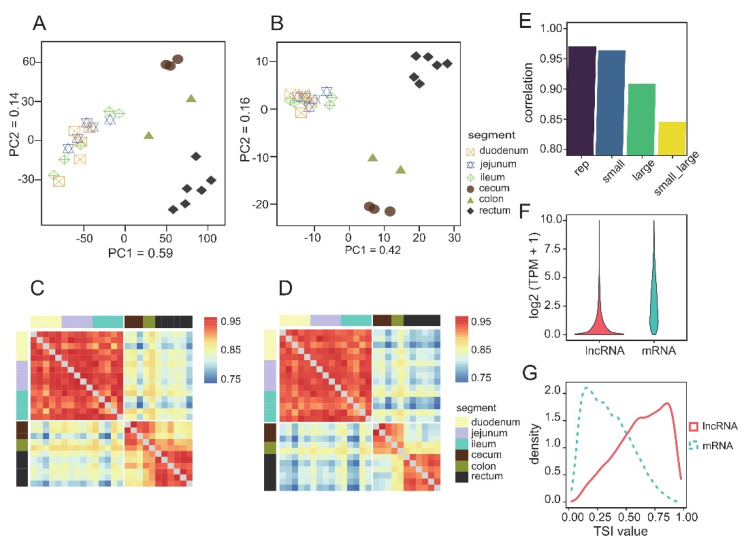
Transcriptome comparison of different intestinal segments. (**A**) PCA of PCGs and (**B**) lncRNAs. The proportion of the variance explained by the principal components (PCs) is indicated in parentheses; (**C**) Heat map showed the correlation of PCGs and (**D**) LncRNAs expression profiles among samples; (**E**) Mean Pearson correlation coefficients within different groups; (**F**) Expression level of PCGs and lncRNAs, TPM value transformed by log2 (TPM + 1); (**G**) Distribution curves of TSI values of PCGs and lncRNAs.

**Figure 2 animals-12-02925-f002:**
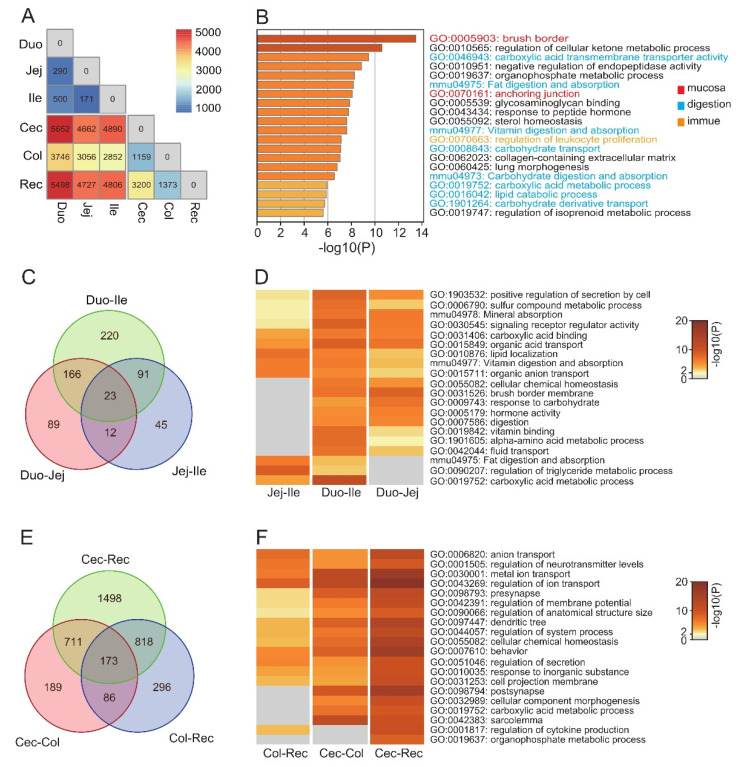
Differential gene analysis between intestinal segments. (**A**) Number of DEGs between paired segments; (**B**) Enrichment analysis of DEGs between small intestine and large intestine, showing the top 20 terms sorted by *p*-value; (**C**) Venn diagrams show the distribution of DEGs in different combinations of small intestine (**D**) and large intestine (**F**); (**E**) Enrichment results of DEGs in different combinations of small intestine. Duo: Duodenum; Jej: jejunum; Ile: ileum; Cec: cecum; Col: colon; Rec: rectum.

**Figure 3 animals-12-02925-f003:**
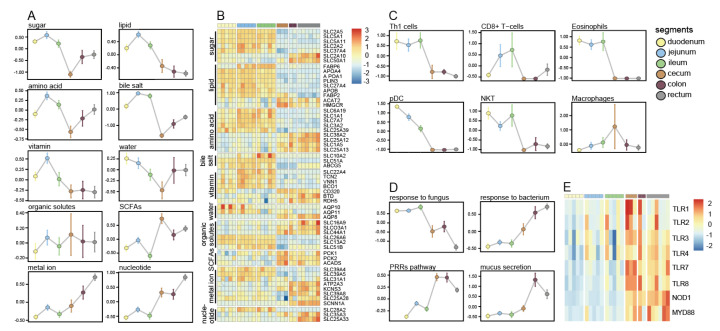
Changes in function among different intestinal segments. (**A**) Expression trends of dige-tion and absorption-related gene sets, the expression levels of genes were normalized; (**B**) Heat map shows the expression levels of genes in different intestinal segments; (**C**) Expression trends of immune-related gene sets; (**D**) Changes in the score of immune cells in the intestine, the results shown in the figure are normalized data; (**E**) The expression levels of immune-related genes.

**Figure 4 animals-12-02925-f004:**
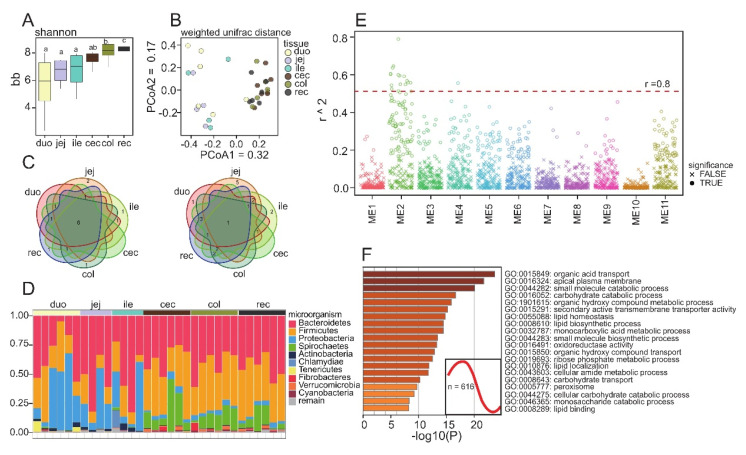
Microorganisms of different intestinal segments (**A**) Alpha diversity analysis based on the faith PD index; (**B**) PcoA analysis based on weighted Unifrac distance; (**C**) Venn diagrams show the distribution of the bacteria in different intestinal segments, and the top 10 most abundant microbes are shown; (**D**) Microbial composition of different intestinal segments at phylum level; (**E**) Manhattan diagram shows the correlation between microorganism and modules, and *p* < 0.01 is regarded as significant correlation. (**F**) Expression trend and functional enrichment results of modules 2. Duo: Duodenum; Jej: jejunum; Ile: ileum; Cec: cecum; Col: colon; Rec: rectum.

**Figure 5 animals-12-02925-f005:**
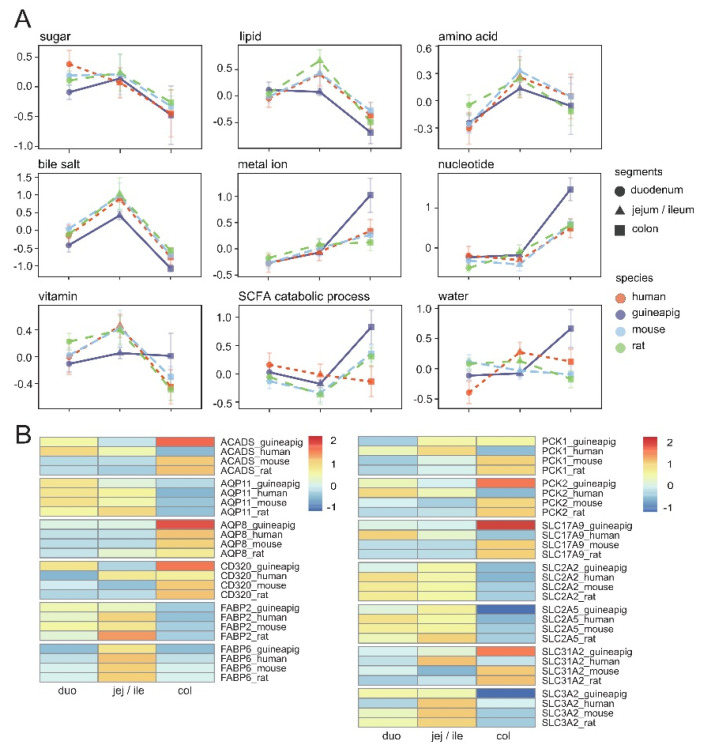
Gene expression differences in intestinal between species (**A**) Expression trend of fun-tional gene sets and (**B**) individual genes in different species. Duo: Duodenum; Jej/Ile: jejunum or ileum; Col: colon.

## Data Availability

The RNA-seq and 16S rRNA seq data are available in the NCBI Sequence Read Archive (SRA; https://www.ncbi.nlm.nih.gov/sra, access date: 1 October 2022) under accession numbers PRJNA832079 and PRJNA832914.

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
