# Peer review of "The Analysis of Transcriptomes and Microorganisms Reveals Differences between the Intestinal Segments of Guinea Pigs"

_animals, 2022, doi:10.3390/ani12212925_

Round 1

Reviewer 1 Report

Dear author

the suggestions are in file

Reviewer 2 Report

The manuscript entitled, "Comprehensive Analysis of Transcriptomes and Microorgan-2 isms Reveals Differences between the Intestinal Segments of 3 Guinea Pigs", compares and contrasts the microbiome population in a group of guinea pigs. Overall, this study is mostly sound other than a few things that I would like clarification on and I believe the interest to the readers is moderate. 

Major concerns:

1. There are a few instances of discussion that appears in the results section such as lines 235-247, 330-337. Please modify the results to reflect the results only and not discussion. 

2. In the paragraph 269-303, there is no reference to a supplementary figure and no discussion of quantitative data, please either add numbers into this paragraph so the readers can determine the differences on their own, add a supplemental figure or reference an existing figure. 

3. For the 16S analysis, were the sample OTUs normalized? If they were not, you would see different results than if they were. 

4. In paragraph 230-247, you discuss the differences between the large and small intestine but I dont see a direct comparison between the small and large intestine in the figure. Perhaps it might be beneficial to include that comparison in Figure 2 if you are going to discuss it in detail. A similar issue occurs in lines 278-303. Also, please move some of this paragraph to the discussion. 

5. In the introduction, please include other studies examining the intestinal microbiota of guinea pigs. 

6. In the abstract (and throughout the paper) you suggest that intestinal transcription and microbial differences came from both the small and large intestine but those are the only options so perhaps you should modify or remove the sentences because they d not make sense. 

7. How did you extract the feces from the tissue? I dont see that discussed in the methods.

8. In lines 393-394 you reference data from other species but I dont see a mention of where that data came from and what their reference numbers are in the methods? If the data is not public than it needs to be deposited in a public database or else this is not an appropriate comparison. 

9. In lines 439-441 you suggest that the previous study made you modify your criteria for the PCGs and lncRNAs but you do not describe the modifications. Please describe here. 

Minor concerns:

1. In line 24, the term variant needs to be modified for clarity

2. In line 25 "to that the" needs to be modified for clarity

3. In line 98, what dies standby mean?

4. In line 112, please include the parameters you used in STAR

5. Please define PCGs and lncRNAs in line 112. 

6. In line 140, please modify "which only one-to-one' for clarity

7. In line 149 please modify "genes that one-to-one" for clarity.

8. In line 180, please define "clean" data

9. Please include the median and range in lines 200 and 201 for the expression profiles

10. Delete one mainly from line 205

11. I do not see a reference for figure 3B.

12. In lines 348-349, please describe what a feature is

13. The sentence in lines 449-450 does not make sense as is. Please rephrase or delete. 

Round 2

Reviewer 1 Report

the paper is ok.

Author Response

Thank you for your comments on our article.

Reviewer 2 Report

The only items that I see in this revision that were insufficiently addressed were the sentences in line 140 (new line 181) and 149 (new line 190; both containing the one-to-one phrase) need to be rephrased as the English tenses are not correct. 

EX: Line 140:  "the genes which only one-to-one ortholog to guinea pig were remained" does not make sense

Line 149: "The genes that one-to-one ortholog to humans and exist in all species are retained" also does not make sense. 

They should be rephrased to "the genes that are only one-to-one orthologs with guinea pig were remained" and "The genes that are one-to-one orthologs to human and exist in all species are retained."

Author Response

Thank you for your careful review of our article, we have revised the two sentences you mentioned.

Reviewer 3 Report

As I mentioned in my first review, the manuscript describes changes in the transcriptome and microorganisms of guinea pig intestine segments. The study was carried out with only six male guinea pigs acquired from a company specializing in experimental animals. The diversity of animal and microorganism gene expression in different tissue segments was investigated by mRNA and 16S RNA sequencing, respectively. The authors describe that the complete intestinal transcription profile and microbial composition of normal guinea pigs provide reference data for intestinal research and the rational use of these experimental animals as models. In addition, they present a comparison of gene expression and microorganisms with other animal species (including humans) with different diets/feeding and environmental situations.

The manuscript has been improved and is much better now. However, I am still concerned about the comparison with other animal species (including humans) that have different diets/feeds and live in different environmental situations. As the authors chose to keep topic 3.5, I suggest that they include a paragraph in the Discussion highlighting the limitations of the study (few sampling, all males from a specific breed of guinea pig, different diets between species, etc.).

Finally, there are still some minor errors that need to be fixed throughout the text. And careful proofreading in English is required.

Author Response

Thank you for your careful review of the article. We have emphasized the shortcomings of this study in the discussion, especially for the cross species analysis (line 567-573), we have also revised other minor errors in the article (line 167-168, line 176-177).

Round 3

Reviewer 3 Report

The authors made the main changes recommended in the article. Additional proofreading is still required for corrections in English writing.